# Efficacy and Safety of a Second Course of Stereotactic Radiation Therapy for Locally Recurrent Brain Metastases: A Systematic Review

**DOI:** 10.3390/cancers13194929

**Published:** 2021-09-30

**Authors:** François Lucia, Ruben Touati, Nicolae Crainic, Gurvan Dissaux, Olivier Pradier, Vincent Bourbonne, Ulrike Schick

**Affiliations:** 1Radiation Oncology Department, University Hospital of Brest, 29200 Brest, France; ruben.touati@chu-brest.fr (R.T.); gurvan.dissaux@chu-brest.fr (G.D.); olivier.pradier@chu-brest.fr (O.P.); vincent.bourbonne@chu-brest.fr (V.B.); ulrike.schick@chu-brest.fr (U.S.); 2Neurology Department, University Hospital of Brest, 29200 Brest, France; nicolae.crainic@chu-brest.fr

**Keywords:** control, local, metastasis, brain, radiosurgery, stereotactic radiotherapy, reirradiation, radionecrosis

## Abstract

**Simple Summary:**

Approximately 30% of patients diagnosed with cancer will ultimately develop brain metastases. Many improvements have been made in systemic and local cancer treatments, which have increased overall survival but also, as a consequence, the number of patients who present with local recurrence following intracranial stereotactic radiotherapy. The management of these recurrences remains controversial. The aim of our review is to evaluate the efficacy and tolerance of a second course of stereotactic radiotherapy.

**Abstract:**

Recent advances in cancer treatments have increased overall survival and consequently, local failures (LFs) after stereotactic radiotherapy/radiosurgery (SRS/SRT) have become more frequent. LF following SRS or SRT may be treated with a second course of SRS (SRS2) or SRT (SRT2). However, there is no consensus on whenever to consider reirradiation. A literature search was conducted according to PRISMA guidelines. Analysis included 13 studies: 329 patients (388 metastases) with a SRS2 and 135 patients (161 metastases) with a SRT2. The 1-year local control rate ranged from 46.5% to 88.3%. Factors leading to poorer LC were histology (melanoma) and lack of prior whole-brain radiation therapy, large tumor size and lower dose at SRS2/SRT2, poorer response at first SRS/SRT, poorer performance status, and no controlled extracranial disease. The rate of radionecrosis (RN) ranged from 2% to 36%. Patients who had a large tumor volume, higher dose and higher value of prescription isodose line at SRS2/SRT2, and large overlap between brain volume irradiated at SRS1/SRT1 and SRS2/SRT2 at doses of 18 and 12 Gy had a higher risk of developing RN. Prospective studies involving a larger number of patients are still needed to determine the best management of patients with local recurrence of brain metastases

## 1. Introduction

It is estimated that nearly 30% of cancer patients will develop brain metastases (BMs) during the clinical course of their illness [1], particularly patients with lung, melanoma, and breast cancer [2]. Alongside systemic treatments and surgery, the role of stereotactic radiosurgery (SRS) or fractionated stereotactic radiotherapy (SRT) for local control (LC) improvement is now well recognized. SRT is now preferred over whole-brain radiotherapy (WBRT) in order to maximize local control while minimizing the probability of neurocognitive decline [3,4], without compromising overall survival (OS). For patients with a limited number of BM, SRS/SRT achieves favorable tumor control rates of 80–90% at 12 months while sparing normal brain tissue, with a radionecrosis (RN) rate of 3 to 5% [5,6].

Recent advances in cancer systemic treatments, like targeted therapies and immune check point inhibitors, have increased OS and consequently, local failures (LFs) after SRS or SRT are observed more frequently [7,8,9]. The purpose of treatment for local recurrence of BM, particularly those that have undergone high-dose irradiation, is to improve the patient’s quality of life by controlling local disease while minimizing the risk of significant toxicity [10]. The management of recurrent metastasis previously treated with SRS or SRT is, however, controversial [10]. Without treatment, the prognosis is poor, with survival often limited to weeks to months, usually due to neurological death [11]. Salvage options include systemic treatment, surgery, RT, or supportive care. Surgical resection may be considered as the preferred salvage modality to obtain histologic confirmation of recurrence and avoid reirradiation [12,13,14,15,16]. However, surgery is difficult if lesions are deeply located or in functional areas and has a significant rate of LF as stand-alone treatment [17]. WBRT alone does not achieve durable LC with a significant risk of neurocognitive sequelae [3]. In these patients, chemotherapy does not provide good results either, with median LC of 2 ± 4 months and OS of 3 ± 7 months [18,19,20]. In contrast, several studies have shown that repeated sessions of SRS/SRT can be performed with very good efficacy and tolerability for the management of new distant BM after a first course of SRS/SRT in order to defer WBRT [21,22]. In the last decade, because of these encouraging results and the lack of effective alternatives, a more frequent use of stereotactic reirradiation (SRS2/SRT2) has been noted in these patients given the major risk of adverse cognitive outcomes after WBRT. However, this approach is arguable both in terms of efficacy and tolerability. First, the effectiveness of a second radiotherapy treatment may be questioned if one considers local recurrence as arising from radio-resistant clones following previous radiotherapy [23]. In addition, SRS2/SRT2 could lead to a major risk of adverse events, most notably symptomatic RN given prior high-dose irradiation of the neighboring central nervous system (CNS) tissue [24]. Thus, it is essential to determine the optimal dose-fractionation scheme for SRS2/SRT2. However, the lack of standardization on SRS and SRT protocols before the International Commission on Radiation Units and Measurements (ICRU) report 91 report made it difficult to compare results between studies performed at different centers [25].

We performed a systematic review of the literature to assess the efficacy and safety of a second course of SRS/SRT treatment in patients with local recurrence of one or more BMs after a first course of SRS/SRT. 

## 2. Materials and Methods

### 2.1. Selection Criteria and Search Strategy

The study was performed according to the Preferred Reporting Items for Systematic Reviews and Meta-Analyzes (PRISMA) guidelines [26]. Because of the lack of a comparator in this setting, the Population, Intervention, Comparison, Outcome (PICO) model was not applicable. Suitable articles were identified by screening 3 electronic databases (Pubmed, Scopus, Cochrane Library) by 2 authors (F.L., R.T.) using the appropriate MeSH terms for the following search items: “brain,” “encephalic,” “cerebral” AND “stereotactic,” “SRS,” “SRT”, “radiosurgery” AND “reirradiation,” “re irradiation,” “salvage,” “repeated,” “repeat” AND “metastases,” “metastatic,” “metastasis,” “in-field relapse,” “in-field progression,” “local relapse,” “local failure”, “recurrent”, and “in-field recurrence”. Only studies evaluating the use of a second stereotactic radiotherapy of BM already irradiated with stereotactic radiotherapy were included in the final analysis. Only fully published, peer-reviewed articles in English, through February 1, 2021, were included in our analysis. Duplicate studies or those with insufficiently reported results were removed. To check compliance with the predetermined research criteria, an assessment was performed independently by 2 authors (V.B., N.C.). In the case of inconsistency or disagreement between the teams, a final decision was made with a third team (U.S., O.P.). The research methodology is shown in Figure 1.

### 2.2. Validity Assessment

For each individual study, risk of bias was assessed using the ROBINS-I tool [27] to estimate the following domains: (a) bias due to confounding of effects; (b) bias in selection of participants; (c) bias due to classification of interventions; (d) bias due to deviations from intended interventions; (e) bias due to missing data; (f) bias in measurement of the outcome; and (g) bias in selection of the reported result. The categories for risk of bias judgements are “low risk”, “moderate risk”, “serious risk”, and “critical risk” of bias. In some cases, no information on which to base a judgement about risk of bias for a domain was available.

### 2.3. Data Extraction and Statistical Analysis

Data extraction from the included articles was performed by one author (R.T.) and data accuracy was checked by a second author (F.L.). Patient, disease, and treatment characteristics, as well as LC rates and incidence of RN, were collected and reported using descriptive statistics. Study-specific 1-year OS values were pooled into a summary value and corresponding 95% confidence intervals (CIs) by applying the “weighted median of medians” (MedCalc Software bvba, Ostend, Belgium; https://www.medcalc.org; accessed on 14 August 2015) method.

When not mentioned explicitly in the text, the 1-year OS was manually retrieved from Kaplan–Meier survival curves (available for all but 1 of the remaining studies). The study-specific proportion of patients with LF at 6 and 12 months of follow-up, or RN at any time after SRS2/SRT2, was pooled into summary proportions and corresponding 95% CIs using random-effects models implemented in Medcalc. Heterogeneity between studies was quantified using the I^2^ statistic, which can be analyzed as the proportion of variability between estimates that is due to true heterogeneity rather than chance [28]. I^2^ values above 50% are considered to denote high heterogeneity. When this occurs, fitting meta-regression and subgroup analysis models (for continuous and categorical variables, respectively) is recommended to find study characteristics that correlate with the outcome and thus may explain some of the heterogeneity.

## 3. Results

The search workflow is outlined in Figure 1. Briefly, 1394 potential references were found. Thirty five full-text papers were initially included after checking for duplicates and pertinence. Among them, 22 were excluded due to a lack of prior SRS/SRT to the target metastasis (previous WBRT only or mixed interventions) or insufficient data reporting [21,28,29,30,31,32,33,34,35,36,37,38,39,40,41,42,43,44,45,46,47,48]. A total of 13 studies met the inclusion criteria and were analyzed in the final meta-analysis [49,50,51,52,53,54,55,56,57,58,59,60,61]. They included 464 patients and 549 treated metastases with a median follow-up of 11 (1–124) months by pooling patients from the 11 studies for which this data was available. 

### 3.1. Validity of Included Studies

All included studies were retrospective. For each study, the risk of bias is reported in Table 1. Despite the presence of possible uncontrolled confounding factors, including differences in primary tumor type and systemic treatments, other local treatment already performed, different radiotherapy protocols at first and second SRS/SRT, and variability in age and performance status, we considered the risk of confounding bias low in all studies. Indeed, in all studies, the clinical and treatment variables were sufficiently precise to allow us to consider the level of risk as non-critical despite the presence of significant heterogeneity in the study population. It is difficult to exclude the possible presence of significant selection bias due to the retrospective nature of all studies. However, this risk is minimized by the use of precise criteria for the differential diagnosis between LF and RN (both at the time of the pre-radiotherapy assessment and in the post-treatment follow-up) in each cohort. All studies provided a precise description of the treatment procedures to avoid major classification bias. In addition, no major deviations from the planned treatment were reported in any study, especially with the short duration of the radiotherapy protocols. Overall, the risk of bias in measurement of the reported results was low thanks to the use of consensual diagnostic criteria, such as the Response Evaluation Criteria in Solid Tumors (RECIST) [53,54,56,57,58], Response Assessment in Neuro-Oncology (RANO)-BM [49,51], and Immunotherapy Response Assessment in Neuro-Oncology (iRANO) [52], or of a validated methodology, such as increased relative cerebral blood volume > 2 (rCBV) extracted from perfusion magnetic resonance imaging (MRI) [50,55,59]. However, the variability in the imaging modality chosen and thus the diagnostic criteria used is of concern. Regarding the risk of bias for missing data, only four studies had short or unreported data, leading to a moderate level of risk. For the remaining articles reviewed, the level of risk was considered low because of the application of an adequate follow-up schedule with sufficient time to assess the efficacy and toxicity of radiation therapy [51,53,56,60]. Nevertheless, no study reported analytical strategies to account for loss of information. This lack of statistical strategies does not allow for concurrent events (death for extracranial progression) to be accounted for, despite the use of a clearly defined estimate of intervention outcome and toxicity, and makes the risk of selective reporting moderate for all studies. Finally, no study had a critical risk of bias, although it is important to note the inter-study variability, particularly with respect to the different modalities of outcome measurement.

### 3.2. Clinical and Treatment-Related Characteristics

Clinical and treatment-related characteristics for all studies are provided in Table 2, Table 3, Table 4, respectively. The median age, available for all studies, was 59 years (range 27–88). Ten studies reported Karnofsky Performance Status (KPS). The majority of patients were in good general condition with a median KPS of 85% (60–100). All studies provided information on the type of primary tumor, but only 9, including 279 patients, provided detailed data with a per patient basis. The most represented histologies were lung cancer (108/279, 38.8%), breast cancer (61/279, 21.9%), melanoma (59/279, 21.1%), and kidney cancer (20/279, 7.2%). The diagnosis of recurrence was mostly made by medical imaging. Only five studies reported pathological evaluation of metastases. It was performed only in 41 out of 121 metastases (33.9%). In the remaining cases, diagnosis of recurrence was based on contrast-enhanced MRI (CE), perfusion MRI, perfusion MRI plus spectroscopy, and perfusion MRI plus DOTA-PET CT in five, two, three, and two studies, respectively. Ten studies reported possible surgical resection of some metastases at any time before SRS2/SRT2. It was performed in 89/458 (19.4%) metastases, but residual disease was systematically present at the time of SRS2/SRT2. All studies reported the eventual possible use of WBRT: WBRT was administered at any time before SRS2/SRT2 in 98/464 (21%). The median dose delivered at the time of the first course of SRS/SRT (SRS1/SRT1) was reported in 8/13 studies with a dose of 20 (18–24) Gy in 1 (1–3) fraction corresponding to a median biological effective dose (BED) of 60 Gy (50.4–81.6) assuming an α/β = 10. The median time between SRS1/SRT1 and SRS2/SRT2 was reported in all studies and was 13 (6–19) months. The median tumor volume at the time of reirradiation was 4.8 (1–40) cc. The median dose delivered at SRS2/SRT2 was 19 (15.5–26.5) Gy in 1 (1–3) fraction corresponding to a median BED of 50.4 (39–70.6) Gy assuming α/β = 10. A single-fraction regimen was used in the majority of cases: 70.7% (388/549). A multiple-fraction regimen was used in 29.3% (161/549) of metastases.

### 3.3. Local Control and Overall Survival

Data for 1-year OS were available in 11 studies. Pooling these results yielded a 1-year OS of 54.2% (95%CI = 38.9–69.1%). However, there was considerable heterogeneity I^2^ = 90.1% (95%CI = 84.3–93.8%) (Figure 2). The only predictive factor found was a lower prescription dose in SRS2/SRT2 significantly associated with poorer survival in one study (Table 4).

In the pooled population of 11 studies, LC at 6 months was 88.4% (95%CI = 85.2–91.2%) with acceptable heterogeneity of I^2^ = 0.0% (95%CI = 0.0–55.9%).

In the pooled population of 12 studies, LC at 1 year was 72.5% (95%CI = 64.6–79.7%) with significant heterogeneity of I^2^ = 74.3% (95%CI = 54.6–85.5%) (Figure 3A). Eight studies reported 2-year LC, 54.2% (95%CI = 42.4–65.8%), with significant heterogeneity I^2^ = 79.0% (CI95 % 58.9–89.2%) (Figure 3B). Six studies reported predictive factors significantly associated with LC. Primary tumor type (melanoma versus other histologies), lack of prior WBRT, large tumor size at SRS2/SRT2, lower dose prescribed at SRS2/SRT2, poorer response at SRS1/SRT1, poorer KPS, and no controlled extracranial disease were factors associated with worse LC (Table 5).

The majority of publications reviewed did not report sufficient information to be included in the meta-analysis.

### 3.4. Radionecrosis

Eleven studies reported the rate of RN, with the cumulative rate yielding a crude median value of 14.3% (95%CI = 8.9–20.6%) in the pooled population at the end of the follow-up period, with significant heterogeneity (I^2^ = 73.3%, (95%CI = 52.5–85.0%)) (Figure 4). However, because no study reported either mean or total follow-up, it was not possible to assess the pooled estimate of incidence. Three studies reported significant risk factors for RN, large tumor volume at the time of SRS2/SRT2, high dose at the time of SRS2/SRT2, large overlap between brain volume irradiated at SRS1/SRT1 and SRS2/SRT2 at doses of 18 and 12 Gy, and a higher value of prescription isodose line at SRS2/SRT2 (Table 4). One study showed a trend toward increased risk with prior WBRT (*p* = 0.05). 

The majority of publications reviewed did not report sufficient information to be included in the meta-analysis.

## 4. Discussion

There are limited data on the best salvage treatment strategy for in-site recurrent BM after initial SRS/SRT. Treatment options include surgical excision, systemic therapy, or re-irradiation with WBRT or SRS2/SRT2. The decision is often guided by a combination of factors including the patients’ age and functional status, control of extracranial disease, intracranial tumor burden, prior treatments, type of primary cancer, and the possibility of targeted therapy [10]. Surgical approach is the preferred treatment option whenever possible to distinguish tumor recurrence from radionecrosis. This strategy provides widely varying results in terms of LC rates, ranging from 62% to 93% at one year [12,13,15] and a median survival of 8.7 months [13]. However, reirradiation is often necessary to achieve better LC even with a neurosurgical approach [63]. Due to its invasiveness associated with a non-negligible risk of mortality and morbidity [12], the use of surgery is currently limited to selected cases, representing 1–11% of patients requiring salvage treatment for recurrent BM [14,64]. Reirradiation of CNS tumors has long been considered not advised because of the belief that normal brain tissue was at risk of irreversible tissue damage. In 1974, Shehata reported on the use of repeated WBRT in patients with progressive breast cancer [65]. However, WBRT increases the risk of subsequent cognitive impairment [66,67] and compromises patient quality of life [3], particularly in long-term survivors who are oligometastatic or with only intracerebral progression. In addition, local recurrence after initial high-dose SRS/SRT treatment is often considered as a radioresistant lesion, so lower doses of WBRT than SRS/SRT are unlikely to achieve long-term disease control. Thus, radiation oncologists remain reluctant to reirradiate the CNS using conventional radiation therapy techniques. Localized irradiation, on the contrary, seems to be an interesting alternative since it may have a more acceptable toxicity profile. One approach to localized irradiation is SRS/SRT as an alternative. The reirradiation of a local target by SRS/SRT is of interest because it would provide better sparing of the healthy CNS compared to WBRT. This strategy has already shown its effectiveness and good tolerance in other pathologies, such as vestibular schwannomas [62] and meningiomas [68]. However, the prescribed doses are lower than those used for BM reirradiation.

In the present review, we investigated the use of a second course of SRS/SRT treatment for recurrent BM. We included 13 retrospective studies with a total of 464 patients and 549 treated metastases. Nine studies reported the histological type of the primary tumor, with a tendency to a higher proportion of melanoma and RCC than the known distribution in case of SRS1/SRT1. One explanation would be that brain metastases of melanoma and RCC are considered radioresistant [69,70]. Indeed, although the response rate to SRS/SRT is encouraging [71], LC seems inferior for these histological types and more particularly for melanoma ranging from 47 to 100% [71,72,73], while those for RCC range from 63 to 100% [71,72,74]. The use of the ROBINS-I tool ensured that there was no critical bias in each domain for all included articles. LC is the primary objective of SRS/SRT, and our meta-analysis found a one-year LC rate of 72.5%, which is close to the results reported by prospective trials on initial SRS/SRT [5,75,76]. Another interesting finding is that the 1-year OS in the pooled population was 54% from the date of SRS2/SRT2. These data are similar to survival after a first course of SRS/SRT in selected subsets of patients with a high graded prognostic assessment (GPA) prognostic index score [74]. These figures are comparable to the results of surgical series in terms of LC [12,13,15] and OS [13]. However, direct comparison between SRS2/SRT2 and resection is limited because the results regarding surgical excision are both for salvage treatment in case of local recurrence but also as a treatment for symptomatic or cortico-dependent RN [16]. Importantly, LC at 1 year and OS at 1 year in the pooled population showed significant heterogeneity between studies, with an I^2^ > 50% for both endpoints. This result may be explained by a large disparity between studies on multiple prognostic factors (radiotherapy regimen, population included, systemic treatment, etc.). However, it may seem surprising that poorer KPS and lack of primary disease control can impact local control. Regarding KPS, one reason could be that these patients were no longer receiving systemic therapy. This information is not available in the relevant study [62]. Concerning the role of primary tumor control, one explanation could be that if the primary tumor is not controlled, it suggests that it is a more aggressive disease and resistant to any treatment and in particular to radiotherapy of brain metastases. Moreover, two studies showed a more important decrease of the LC at 2 years [50,51]. These results may be explained by the larger volumes of the treated metastases [50,51], which may have led to a lower median margin dose BED [50]. Unreported prognostic factors (number lost to follow-up, systemic treatment, and dosimetric parameters) could also explain this result. Another major finding of our review is the pooled rate of RN, estimated at 13% at a median follow-up of 11 months. Previous studies have found an increased risk of RN after SRS2/SRT2 of BM [24,55]. However, it should be noted that the incidence of RN is variable among the different salvage modalities of rescue therapy for recurrent BM after SRS/SRT. Rae et al. [64] investigated the risk of RN based on the salvage treatment type in patients with recurrent BM after SRS/SRT, excluding SRS2/SRT2. There was no significant increase in the incidence of RN in patients receiving salvage therapy, as the baseline rate of RN was 4.5% in patients with salvage therapy (regardless of salvage strategy) but increased to 21% in patients receiving simultaneous distant-site SRS/SRT and WBRT. Three studies (see Table 4) found three predictive factors significantly correlated with an increased risk of RN: (i) higher prescribed dose at the time of SRS2/SRT2 (two studies), (ii) larger tumor volume at the time of SRS2/SRT2 (two studies), and (iii) a larger area of overlap between the volumes of SRS1/SRT1 and SRS2/SRT2 (one study). A possible way to decrease the risk of toxicity with reirradiation is the use of fractionated stereotactic radiotherapy. In this case, the treatment is typically delivered in three to five sessions. Indeed, CNS tissues are known to be very sensitive to dose per fraction, so a small change in fractionation can potentially reduce the risk of RN [75,76]. This approach is being evaluated in the initial treatment of large BM [76,77]. The role of fractionation in the course of reirradiation requires further study. In addition, there are no evidence-based OAR dose constraints. Indeed, multiple recommendations exist including American Association of Physicists in Medicine Task Group (AAPM-TG) 101 [78], Timmerman constraints [79] (updated in 2017 [80]), and the United Kingdom (UK) Consensus Guidelines [81] and HYTEC [82], but all of these recommendations have differences. This results in a large variability in the dose constraints used in the current trials, limiting the interpretation of toxicity rates or the acceptability of the dose constraints used [82].

The main limitations of our work are mainly its retrospective nature and the small number of eligible studies meeting our search criteria. In addition, there are different validated parameters for the differential diagnosis between progression and RN and the definition of response after SRS2/SRT2, but there is no standardized definition of these events. Thus, in the different series, different parameters were used, which represents a potential source of bias. This should be corrected with the consensus use of objective criteria, such as RANO-BM [83]. However, in cases of recurrent BM after salvage reirradiation, the diagnosis of LF and RN is extremely difficult [40]. Despite numerous diagnostic methods, including MR spectroscopy, perfusion [84,85], and methionine positron emission tomography [86], the diagnosis remains difficult because of the overlapping radiographic appearance. Studies are ongoing to determine the best diagnostic strategy for differential diagnosis between radiation necrosis and BM relapse (NCT02636634; NCT04111588; NCT04244019). Finally, the lack of detailed reporting on multiple prognostic factors makes it difficult to determine their potential impact on observed outcomes. This mainly concerns systemic treatments and the great variability in the prescription of radiotherapy and its dose reports. Indeed, systemic treatments may have high intracranial efficacy, including next-generation tyrosine kinase inhibitors as well as immune checkpoint inhibitors, particularly in patients with asymptomatic disease for lung cancer, breast cancer, or melanoma [87]. Moreover, no studies consider the additional effect of systemic therapy. However, the authors considered that medical oncologists would not change the systemic treatment or start a new treatment after identifying a locally recurrent BM. In addition, they considered that pursuing the same systemic treatment had no effect on the outcome of SRS2/SRT2 BM because a local recurrence during this treatment can be a sign of an acquired drug resistance. The great variability of dose prescription (fractionation, prescription isodose, biological equivalent, etc.) in SRS/SRT was highlighted in the ICRU91 report [25]. It aimed to standardize the modalities of dose reporting in SRS/SRT, but it was published after the majority of the studies included in our review. Values of I^2^ above 50% for different outcomes are considered as denoting large heterogeneity. Thus, meta-regression and subgroup analysis models (for continuous and categorical variables, respectively) should have been performed to find study characteristics that might explain some of the heterogeneity. However, because of the lack of detail on the parameters that might be associated with the outcomes, these analyses could not be conducted.

Because of these different elements, no conclusions can be drawn on patient selection, neither the efficacy of SRS2/SRT2 nor the risk of RN.

Future studies should explore new strategies for recurrent BM, including different radiotherapy strategies (SRS or SRT), or treatments, such as the use of laser-induced thermotherapy [88] or the combination of several treatments (e.g., SRS/SRT and new systemic therapies), to reduce the risk of RN while improving local effectiveness. One study is currently investigating the combination of laser interstitial thermotherapy and pembrolizumab (NCT04187872).

## 5. Conclusions

A second SRS/SRT treatment (SRS2/SRT2) is an effective strategy for local recurrence of BM after initial SRS/SRT treatment, with similar results to surgical series in terms of overall survival and LC. However, SRS2/SRT2 results in symptomatic RN in 13% of cases, a severe adverse event that can cause significant impairment of quality of life with physical or psychological impact. Unfortunately, as the available data are very heterogeneous and scarce, no reliable analysis looking for predictive factors for efficacy or toxicity could be performed. ICRU 91 standardization of dose ratios for stereotactic treatments will allow for better comparison of different protocols and their outcomes. Standardization of the criteria for differential diagnosis between local recurrence and RN seems necessary. Finally, real consideration should be given to the need for a second course of SRS in asymptomatic BM if high-efficacy intracranial systemic therapy is possible. Prospective studies involving a larger number of patients are still needed to determine the best management of patients with local recurrence of BM.

## Figures and Tables

**Figure 1 cancers-13-04929-f001:**
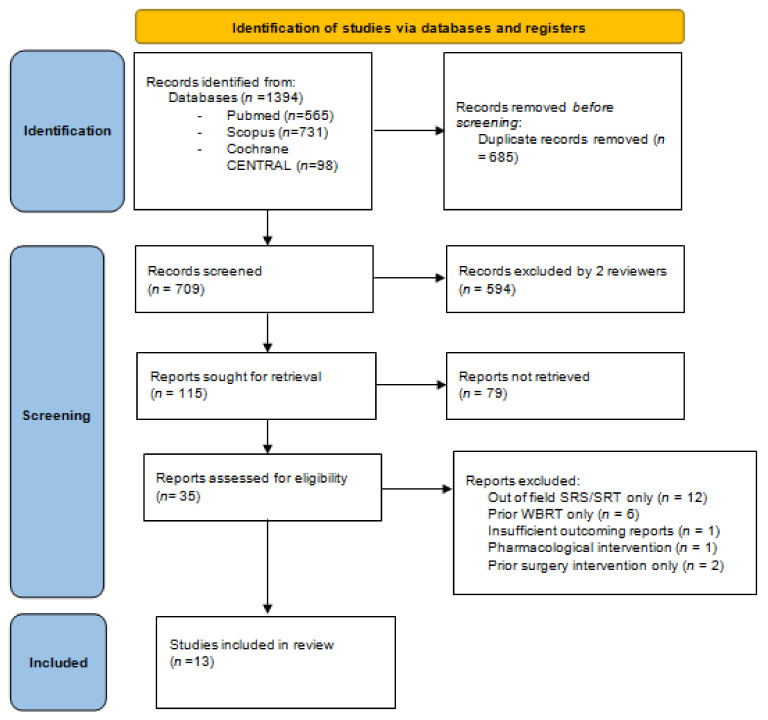
PRISMA 2020 flow diagram for our systematic review, which included searches of databases only.

**Figure 2 cancers-13-04929-f002:**
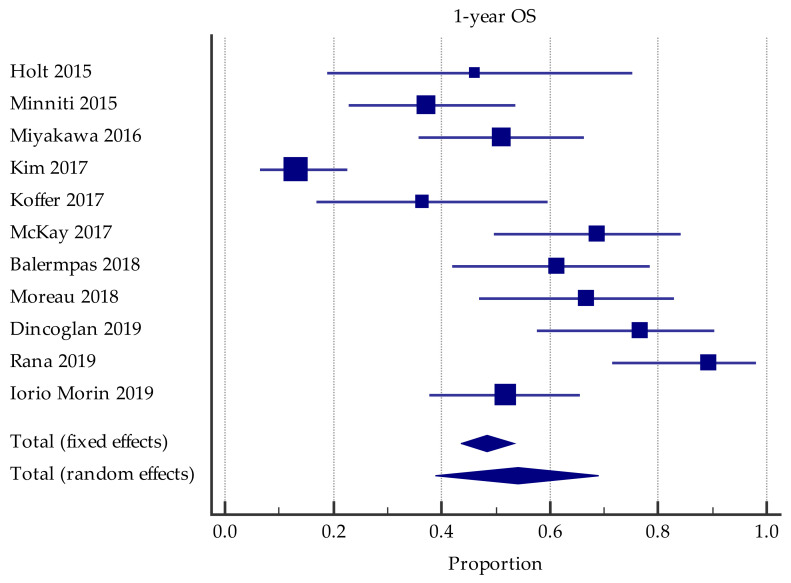
Forest plot of studies evaluating SRS2/SRT2: effect on 1-year overall survival.

**Figure 3 cancers-13-04929-f003:**
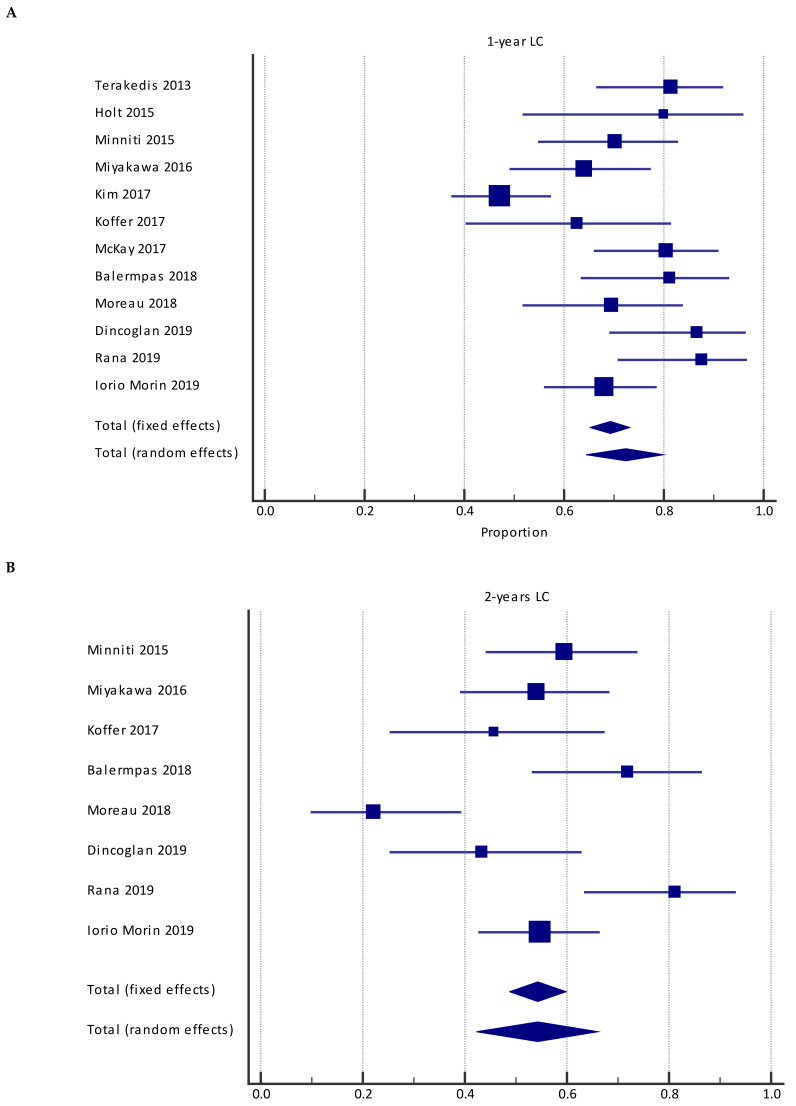
Forest plot of studies evaluating SRS2/SRT2: effect on 1-year local control (**A**) and 2-year local control (**B**).

**Figure 4 cancers-13-04929-f004:**
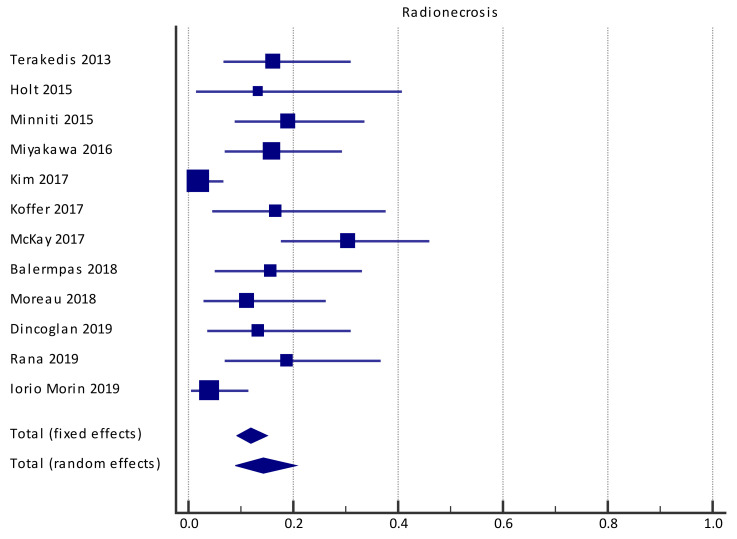
Forest plot of studies evaluating SRS2/SRT2: effect on radionecrosis.

**Table 1 cancers-13-04929-t001:** Risk of bias.

Study	Confounding of Effects	Selection of Participants	Classification of Interventions	Deviations from Intended Interventions	Missing Data	Measurement of the Outcome	Selection of the Reported Result
Terakedis 2013 [57]	+	+	+	+	+	+	++
Greto 2014 [56]	+	+	+	+	++	+	++
Holt 2015 [58]	+	+	+	+	+	++	++
Minniti 2015 [55]	+	+	+	+	+	+	++
Miyakawa 2016 [61]	+	+	+	+	+	++	++
Kim 2017 [60]	+	+	+	+	++	+	++
Koffer 2017 [59]	+	+	+	+	+	+	++
McKay 2017 [54]	+	+	+	+	+	+	++
Balermpas 2018 [52]	+	+	+	+	+	+	++
Moreau 2018 [51]	+	+	+	+	++	+	++
Dincoglan 2019 [50]	+	+	+	+	+	+	++
Rana 2019 [53]	+	+	+	+	++	+	++
Iorio Morin 2019 [62]	+	+	+	+	+	+	++

Risk of bias according the ROBINS-I tool; (+): low risk; (++): moderate risk.

**Table 2 cancers-13-04929-t002:** Summary of clinical characteristics.

Study	Number of Patients	Number of BM	Median Age (Range)	Male	Female	Median KPS (Range)	Lung Tumor	Breast Tumor	RCC	Melanoma	Other	Prior Surgery	Systemic Treatment Before	Median Number of Metastases	Metastases Biopsy Available	Modality Diagnostic	Criteria Diagnostic	Median Follow-Up
Terakedis 2013 [57]	37	43	51 (27–84)	13	14	NA	9	8	2	20	4	13	NA	1	NA	MRI	RECIST	7 (1–45)
Greto 2014 [56]	11	11	47 (33–77)	NA	NA	80 (60–100)	4	3	NA	NA	4	NA	11	1	NA	MRI	RECIST	4 (1–7)
Holt 2015 [58]	13	15	53 (30–70)	5	8	80 (70–90)	1	2	1	9	2	15	NA	1	15	NA	NA	9 (2.2–54.9)
Minniti 2015 [55]	43	47	61	21	22	80 (60–100)	17	9	NA	11	6	NA	NA	1	NA	perfusion MRI and FDOPA PET	rCBV > 2	19 (2–27)
Miyakawa 2016 [61]	47	50	61 (40–85)	20	27	> 70	20	13	7	NA	7	0	NA	1	7	MRI + PET methionine ou biopsie	NA	10 (1–40)
Kim 2017 [60]	84	108	59.4 (mean)	68	46	NA	79	10	15	NA	10	0	NA	1	NA	MRI spectroscopy, perfusion & TEP	RECIST	NA
Koffer 2017 [59]	22	24	59 (43–80)	6 (37%)	16 (64%)	NA	9	2	2	0	9	5	NA	1	5	MRI perfusion, spectroscopy, PET	rCBV > 2 and cho/cr	8.8 (NA)
McKay 2017 [54]	32	46	59 (36–88)	NA	NA	80 (60–100)	16	9	2	2	3	23	NA	1	11	perfusion MRI	RECIST	24 (12–124)
Balermpas 2018 [52]	31	32	65 (43–81)	15	16	90	10	10	1	5	5	9	14	1	NA	MRI	iRANO	12 (1–66)
Moreau 2018 [51]	30	36	59 (39–83)	20	10	90–100 (70–100)	15	5	0	4	6	3	NA	1	3	perfusion MRI	RANO-BM	14 (1–107)
Dincoglan 2019 [50]	30	30	57 (38–78)	16	14	80 (70–100)	11	9	3	4	3	NA	NA	1	NA	MRI perfusion, spectroscopy, PET	rCBV > 2 cho/cr	22 (10–45)
Rana 2019 [53]	28	32	60 (NA)	17	11	80–100 (70–100)	3	5	5	11	4	9	15	1	NA	MRI	RECIST	NA
Iorio Morin 2019 [62]	56	75	57 (27–81)	20	36	90	34	11	4	4	3	12	54	4	NA	MRI perfusion, spectroscopy, PET	RANO-BM	11

Abbreviations: BM = brain metastases; KPS = Karnofsky performance status scale; RCC = renal cell carcinoma; NA = not available; MRI = magnetic resonance imaging; RECIST = response evaluation criteria in solid tumors; FDOPA PET = 3,4-dihydroxy-6-^18^F-fluoro-phenylalanine positron emission tomography; rCBV = relative cerebral blood volume; cho/cr = choline-to-creatine ratio; iRANO = immunotherapy response assessment in neuro-oncology; RANO-BM = response assessment in neuro-oncology-brain metastases.

**Table 3 cancers-13-04929-t003:** SRS1/SRT1 treatment characteristics.

Study	WBRT	Number of SRS1	Number of SRT1	Median Volume cc	Median Isodose Line (%)	Median Maximum Dose SRS/SRT1	Median Maximum Dose BED SRS/SRT1	Median Margin Dose SRS/SRT1	Median Margin Dose BED SRS/SRT1
Terakedis 2013 [57]	17	43	0	NA	95	18.94	53.05	18	50.4
Greto 2014 [56]	6	2	11	NA	80 (70–100)	NA	NA	NA	NA
Holt 2015 [58]	1	13	0	4.3 (0.76–19.3)	80	26.3 (22.5–33.8)	81.4 (73.5–124.9)	21 (18–27)	65.1 (58.8–99.9)
Minniti 2015 [55]	0	47	0	NA	NA	NA	NA	NA	NA
Miyakawa 2016 [61]	0	47	0	10.4 (0.4–72.5)	50	40	120	20	60 Gy
Kim 2017 [60]	0	108	0	3.89 (0.024–25.5)	50	37.8 (24–48)	109.24	18.9 (12–24)	54.6
Koffer 2017 [59]	8	24	0	2.25	NA	NA	NA	18 (17–20)	50.4 (45.9–60)
McKay 2017 [54]	8	46	0	1.28 (0.01–22.6)	NA	NA	NA	20 (12–24)	60 (26.4–81.6)
Balermpas 2018 [52]	5	30	2	2.0 (0.1–22.9)	65 (32–78)	29.5 (22.1–44.0)	110 (70.7–237.6)	23.8 (18.0–31.1)	79.3 (50.5–113.9)
Moreau 2018 [51]	24	NA	NA	NA	NA	NA	NA	NA	NA
Dincoglan 2019 [50]	0	30	0	8.85 (0.1–21.6)	85–95	20	56	18 (16–24 Gy)	50.4 Gy
Rana 2019 [53]	8	30	2	0.48 (0.02–6.70)	83.5 (69–96)	28.74	97.72	24 (18–30)	81.6
Iorio Morin 2019 [62]	21	75	0	0.86 (0.01–27.3)	50 (45–85)	40 (28–48)	120 (67.2–163.2)	20 (14–24)	60 (33.6–81.6)

Abbreviations: WBRT = whole brain radiation therapy; SRS1 = first course of stereotactic radiosurgery; SRT1 = first course of fractionated stereotactic radiotherapy; NA = not available; BED = biological effective dose.

**Table 4 cancers-13-04929-t004:** SRS2/SRT2 treatment characteristics.

Study	Median Delay SRS1/SRT1 to SRS2/SRT2 (in Months)	Number of SRS2	Number of SRT2	Median Volume cc	Median Isodose Line (%)	Median Maximum Dose SRS/SRT2	Median Maximum Dose BED SRS/SRT2	Median Margin Dose SRS/SRT2	Median Margin Dose BED SRS/SRT2
Terakedis 2013 [57]	9	43	0	1.5	95	18.9	53.1	18	50.4
Greto 2014 [56]	13 (4–34)	7	4	40.43 (7–374)	80 (70–80)	24.375	71.9	19.5 (12–30)	57.53
Holt 2015 [58]	6.4 (2.4–15.2)	6	9	9.4 (0.57–23)	80	26.25 (20–37.5)	73.5	21 (16–30)	58.8
Minniti 2015 [55]	17 (6–56)	0	47	12.3 (1.5–33.1)	85 (80–90)	50.8 (42–50.8)	35.7 (35.7–43.2)	24 (21–24)	43.2 (35.7–43.2)
Miyakawa 2016 [61]	7.5 (1–33)	0	50	28.8 (7.1–103)	90	33.3	43.3	30	39
Kim 2017 [60]	9.1 (2.5–58.3)	108	0	5.94 (0.42–29.9)	50%	34 (12–48)	92.2	17 (12–24)	46.08
Koffer 2017 [59]	13.4 (1.9–52.4)	24	0	3.3	NA	NA	NA	15.5 (10–20)	39.53 (20–60)
McKay 2017 [54]	19 (2–98)	46	0	0.98 (0.01–19.7)	NA	NA	NA	20 (14–22)	60 (33.6–70.4)
Balermpas 2018 [52]	12.4 (3.2–88.2)	24	8	2.5 (0.1–37.5)	69 (53–80)	28 (17.4–38.1)	97.2 (40.1–126.3)	23.5 (14.3–33)	70.6 (34.5–89.9)
Moreau 2018 [51]	15.4 (11-78)	36	0	4.8 (0.13–24.8)	90	20	45.36	18 (12–20)	50.4
Dincoglan 2019 [50]	13.5 (3.7–49)	0	30	14.6 (1.6–35.6)	85-95	23.33	39.7	21 (21–30)	35.7 (35.7–48)
Rana 2019 [53]	9.7 (2.5–56.9)	19	13	1.35 (0.11–34.9)	83.5 (69–96)	31.73	77.96	26.5 (18–36)	65.1
Iorio Morin 2019 [62]	13 (3-47)	75	0	1.19 (0.07–20.6)	50 (30–80)	100.8 (52.8–120)	36 (24–40)	18 (12–20)	50.4 (26.4–60)

Abbreviations: SRS1 = first course of stereotactic radiosurgery; SRT1 = first course of fractionated stereotactic radiotherapy; SRS2 = second course of stereotactic radiosurgery; SRT2 = second course of fractionated stereotactic radiotherapy; NA = not available; BED = biological effective dose.

**Table 5 cancers-13-04929-t005:** Oncological endpoints and toxicities.

Study	1-Year OS (%)	2-Years OS (%)	6 Months LC (%)	1-Year LC (%)	2-Years LC (%)	Toxicity Any Grade (%)	Toxicity > 2 (%)	RN Radiological (%)	Variables Related to OS [HR(95%CI)]	Variables Related to LC [HR(95%CI)]	Variables Related to RN [HR(95%CI)]
Terakedis 2013 [57]	NA	NA	83.3	80.6	NA	NA	NA	16	NA	no	NA
Greto 2014 [56]	NA	NA	NA	NA	NA	15.4	0	NA	NA	NA	NA
Holt 2015 [58]	43.8 (8.6–59.4)	NA	100	75 (31.5–93.1)	NA	15.4	15	15	No	No	No
Minniti 2015 [55]	37	20	91	70	60	55	17	19	NA	Melanoma [7.1 (1.9–21)]	-Overlap V18_SRS1/SRT1_/V12_SRS2/SRT2_>10cc [3.1 (1.1–13.6)]
Miyakawa 2016 [61]	50	22	85	63	54	49	NA	17	NA	NA	NA
Kim 2017 [60]	13.2	NA	NA	46.5	NA	NA	NA	1.8	No	-prescription radiation dose of 16 Gy (*p* = 0.000); -tumor volume less than both 4 mL (*p* = 0.001) and 10 mL at SRS2/SRT2 (*p* = 0.008)	NA
Koffer 2017 [59]	37.5	17.5	94.1	61.1	48	NA	NA	16	No	PTV size SRS2/SRT2 > 4cc [NA(NA)]	No (Trend for prior WBRT, *p* = 0.05)
McKay 2017 [54]	70 (55–88)	0	90	79 (67–94)	NA	35	11	36	Lower dose level SRS2/SRT2 [0.64 (0.49–0.84)]	no	-Tumor volume SRS2/SRT2 [1.19 (1.07–1.32)] -Dose SRS2/SRT2 [0.64 (0.48–0.84)] NB V40_SRS/SRT1+2_>0.76cc = 20% NTCP
Balermpas 2018 [52]	61.7	46.3	92	79.5	71.5	19.4	12.9	16.1	No	no	No
Moreau 2018 [51]	65.5 (47.3–80)	27.6 (14.7–45.7)	82.9 (67.6–91.9)	67.8 (51–81)	22	36	0	10	No	Prior WBRT [0.25 (0.1–0.64)]; PTV < 3cc [0.19 (0.1–0.52)]	no
Dincoglan 2019 [50]	76	34.9	93	86	44	NA	3.3	13	NA	PTV size SRS2/SRT2 > 20cc [NA(NA)]	No
Rana 2019 [53]	90.6 (79–100)	48.6 (28.4–83.3)	90	88.3 (76.7–100)	80.3 (63.5–100)	NA	NA	18.8	NA	no	-Higher prescribed IDL [HR0.886 (0.788–0.995)] -Tumor Volume SRS2/SRT2 > 0.48cc [1.55 (1.05–2.29)]
Iorio Morin 2019 [62]	52	37	85	68	55	5.0	1.3	4.0	NA	Higher Dose SRS2/SRT2 [0.79(0.69–0.90)]; Best Response SRS1/SRT1 • CR [0.026 (0.003–0.24)] • PR [0.062 (0.008–0.46)] • SD [0.090 (0.012–0.64)] -KPS SRS2/SRT2 [0.93 (0.88–0.99)]; Active primary tumor [0.15 (0.049–0.48)]	no

Abbreviations: OS = overall survival; LC = local control; RN = radionecrosis; HR = hazard ratio; CI = confidence interval; NA = not available; SRS1 = first course of stereotactic radiosurgery; SRT1 = first course of fractionated stereotactic radiotherapy; SRS2 = second course of stereotactic radiosurgery; SRT2 = second course of fractionated stereotactic radiotherapy; WBRT = whole brain radiation therapy; PTV = planning target volume; CR = complete response; PR = partial response; SD = stable disease; KPS = Karnofsky performance status scale; IDL = isodose line; V_x_ = percentage of CNS receiving x Gy or higher.

## Data Availability

Not applicable.

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
