# Peer review of "Efficacy and Safety of a Second Course of Stereotactic Radiation Therapy for Locally Recurrent Brain Metastases: A Systematic Review"

_cancers, 2021, doi:10.3390/cancers13194929_

Round 1
Reviewer 1 Report
Comments:
This is a very well written manuscript and excellent analysis of a topic difficult to ascertain from the published literature. The lack of clear reporting outcomes is highlighted by the authors and despite this, they perform a detailed and well outlined analysis of pooled outcomes. Two minor comments:
- Please remove underscore of “Pubmed” in Figure 1.
- Line 150 “The” should not be capitalized
Author Response
Comments and Suggestions for Authors
Comments:
This is a very well written manuscript and excellent analysis of a topic difficult to ascertain from the published literature. The lack of clear reporting outcomes is highlighted by the authors and despite this, they perform a detailed and well outlined analysis of pooled outcomes. Two minor comments:
- Please remove underscore of “Pubmed” in Figure 1.
Sorry, this has been done.
- Line 150 “The” should not be capitalized
Sorry, this has been done.
Reviewer 2 Report
How to provide salvage therapy for local recurrence after SRS/SRT of BM is an important and clinically relevant question in these days when the prognosis of patients with BM continues to improve by newly developed systemic treatment. This study focused on SRS/SRT as one of the answers to this clinical question, and tried to answer it objectively to the maximum extent possible by reviewing the existing evidence. On the other hand, the limitations of the approach using the meta-analysis were appreciated as well. There are no major changes that should be made, but some questions and suggestions from this reviewer are listed below.
Minor points:
Line 178-.
"The most represented histologies were lung cancer 178 (108/279, 38.8%), breast cancer (61/279, 21.9%), melanoma (59/279, 21.1%), and kidney 179 cancer (20/279, 7.2%). "
This is clearly different from the general distribution of primary cancers in BM, which seems to have a higher proportion of melanoma and kidney cancer, known as radioresistant tumors. This reviewer recommends providing the authors' perspective on this point in the Discussion section.
Line 219
"heterogeneity of I2=0.0% (CI95% 0.0-55.9%)."
Just to confirm, is 0% really correct?
Line 226.
"poorer KPS, and no controlled extracranial disease"
This reviewer could not understand the reason that poorer KPS, and no controlled extracranial disease had a negative impact on local control. This reviewer recommends providing the authors' perspective on this point in the Discussion section.
Line 228.
"The majority of publications reviewed did not report sufficient information to be included in the meta-analysis."
At the end of the subsection "Local control and overall survival", this reviewer could not understand the meaning of this phrase.
Line 281
" One approach to localized irradiation is intraoperative radiotherapy. Shibamoto et al [67] achieved durable remission after tumor resection and intraoperative radiation in patients with anaplastic ependymoma or anaplastic oligodendroglioma."
This reviewer felt this part to be a bit too abrupt and irrelevant. The reference 67 is a study of malignant gliomas, not BM, so this reviewer suggests deleting this part.
Line 301.
"a symptomatic treatment".
This reviewer could not understand the meaning of this sentence, so it should be reworded to make it easier to understand.
Tables & Figures
It would be easier to read if a reference number was provided to each study’s shoulder.
In Forrest plots, were the studies listed in alphabetical order? If so, Iorio Morin was at the bottom of the list and it should be corrected. This reviewer thought the order by publication year would be more desirable than alphabetical order.
Fig.3A
It would be better to add “1-year LC” at the top. Similarly, should “1-year OS” be added to the top of Figure 2 for consistency?
Table 3
There was a duplicate column "median volume cc".
References
References were numbered up to 86 in the body text, but there were only 81 references in the reference list. Please add the appropriate references to the list.
Author Response
Comments and Suggestions for Authors
How to provide salvage therapy for local recurrence after SRS/SRT of BM is an important and clinically relevant question in these days when the prognosis of patients with BM continues to improve by newly developed systemic treatment. This study focused on SRS/SRT as one of the answers to this clinical question, and tried to answer it objectively to the maximum extent possible by reviewing the existing evidence. On the other hand, the limitations of the approach using the meta-analysis were appreciated as well. There are no major changes that should be made, but some questions and suggestions from this reviewer are listed below.
Minor points:
- Line 178-.
"The most represented histologies were lung cancer 178 (108/279, 38.8%), breast cancer (61/279, 21.9%), melanoma (59/279, 21.1%), and kidney 179 cancer (20/279, 7.2%). "
This is clearly different from the general distribution of primary cancers in BM, which seems to have a higher proportion of melanoma and kidney cancer, known as radioresistant tumors. This reviewer recommends providing the authors' perspective on this point in the Discussion section.
We have added this part in the discussion
“Nine studies reported the histological type of the primary tumor, with a tendency to a higher proportion of melanoma and RCC than the known distribution in case of SRS1/SRT1. One explanation would be that brain metastases of melanoma and RCC are considered radioresistant [69, 70]. Indeed, although the response rate to SRS/SRT is encouraging [71], LC seems inferior for these histological types and more particularly for melanoma ranging from 47 to 100% [71-73], while those for RCC range from 63 to 100% [71, 72, 74].”
- Line 219
"heterogeneity of I2=0.0% (CI95% 0.0-55.9%)."
Just to confirm, is 0% really correct?
Yes, we confirm this result.
- Line 226.
"poorer KPS, and no controlled extracranial disease"
This reviewer could not understand the reason that poorer KPS, and no controlled extracranial disease had a negative impact on local control. This reviewer recommends providing the authors' perspective on this point in the Discussion section.
We have added this part in the discussion
“However, it may seem surprising that poorer KPS and lack of primary disease control can impact local control. Regarding KPS, one reason could be that these patients were no longer receiving systemic therapy. This information is not available in the relevant study [62]. Concerning the role of primary tumor control, one explanation could be that if the primary tumor is not controlled, it suggests that it is a more aggressive disease and resistant to any treatment and in particular to radiotherapy of brain metastases.”
- Line 228.
"The majority of publications reviewed did not report sufficient information to be included in the meta-analysis."
At the end of the subsection "Local control and overall survival", this reviewer could not understand the meaning of this phrase.
We meant that we did not have sufficiently detailed information on possible prognostic factors (e.g., systemic therapy or dosimetric parameters of radiotherapy) for each patient in each study to perform a meta-analysis.
- Line 281
"One approach to localized irradiation is intraoperative radiotherapy. Shibamoto et al [67] achieved durable remission after tumor resection and intraoperative radiation in patients with anaplastic ependymoma or anaplastic oligodendroglioma."
This reviewer felt this part to be a bit too abrupt and irrelevant. The reference 67 is a study of malignant gliomas, not BM, so this reviewer suggests deleting this part.
This part has removed.
- Line 301.
"a symptomatic treatment".
This reviewer could not understand the meaning of this sentence, so it should be reworded to make it easier to understand.
We reworded this sentence
“…also as a treatment for symptomatic or cortico-dependent RN”
Tables & Figures
- It would be easier to read if a reference number was provided to each study’s shoulder.
This has been done in each table.
- In Forrest plots, were the studies listed in alphabetical order? If so, Iorio Morin was at the bottom of the list and it should be corrected. This reviewer thought the order by publication year would be more desirable than alphabetical order.
Sorry, this has been done.
- 3A
It would be better to add “1-year LC” at the top. Similarly, should “1-year OS” be added to the top of Figure 2 for consistency?
Sorry, this has been done.
- Table 3
There was a duplicate column "median volume cc".
Sorry, this has been done.
- References
References were numbered up to 86 in the body text, but there were only 81 references in the reference list. Please add the appropriate references to the list.
Sorry, this has been done.
Reviewer 3 Report
Lucia et al. performed a systematic review of the literature to assess the efficacy and safety of the second course of SRS/SRT treatment in patients with local recurrence of one or more BMs after the first course of SRS/SRT. Since there are limited data on the best salvage treatment strategy for in-site recurrent BM after initial SRS/SRT, this review is of interest. They included 13 retrospective studies with 464 patients and 549 treated metastases. Second SRS/SRT treatment (SRS2/SRT2) gave similar results to surgical series in terms of overall survival and LC for local recurrence of BM after initial SRS/SRT treatment. They also identified three predictive factors significantly correlated with an increased risk of RN: i) higher prescribed dose at the time of SRS2/SRT2 ii) larger tumor volume at the time of SRS2/SRT2 iii) and a larger area of overlap between the volumes of SRS1/SRT1 and SRS2/SRT2. The main limitations of this work are its retrospective nature and the small number of eligible studies. In addition, different parameters were used for the differential diagnosis between progression and RN and the definition of response after SRS2/SRT2, which represents a potential source of bias. Moreover, the available data are very heterogeneous and a more stringent standardization of the criteria used should be applied.
Authors reported in Figure 3 the effects of SRS2/SRT2 on 1-year and 2-years local control. How do authors explain that for the studies of Dincoglan 2019 and Moreau 2018, a more important decrease of the LC at 2 years was observed? Please comment.
Table 1 concerning the risk of bias is very subjective. At least, authors must clarify the code used for the classification of the risk of bias.
As a general comment, the tables are sometimes very difficult to read and interpret compared with the main text of the manuscript. For example, the authors talked about 279 patients selected in nine studies l.177 with the histologies. However, when we look at the table, the information concerning biopsies is given for five biopsies. Please clarify.
Author Response
Comments and Suggestions for Authors
Lucia et al. performed a systematic review of the literature to assess the efficacy and safety of the second course of SRS/SRT treatment in patients with local recurrence of one or more BMs after the first course of SRS/SRT. Since there are limited data on the best salvage treatment strategy for in-site recurrent BM after initial SRS/SRT, this review is of interest. They included 13 retrospective studies with 464 patients and 549 treated metastases. Second SRS/SRT treatment (SRS2/SRT2) gave similar results to surgical series in terms of overall survival and LC for local recurrence of BM after initial SRS/SRT treatment. They also identified three predictive factors significantly correlated with an increased risk of RN: i) higher prescribed dose at the time of SRS2/SRT2 ii) larger tumor volume at the time of SRS2/SRT2 iii) and a larger area of overlap between the volumes of SRS1/SRT1 and SRS2/SRT2. The main limitations of this work are its retrospective nature and the small number of eligible studies. In addition, different parameters were used for the differential diagnosis between progression and RN and the definition of response after SRS2/SRT2, which represents a potential source of bias. Moreover, the available data are very heterogeneous and a more stringent standardization of the criteria used should be applied.
We agree with these limitations, which can be explained by the retrospective nature of all studies
- Authors reported in Figure 3 the effects of SRS2/SRT2 on 1-year and 2-years local control. How do authors explain that for the studies of Dincoglan 2019 and Moreau 2018, a more important decrease of the LC at 2 years was observed? Please comment.
Indeed, we agree with this comment.
We have added this part in the discussion.
“Moreover, two studies showed a more important decrease of the LC at 2 years [50, 51]. These results may be explained by larger volumes of treated metastases [50, 51] that may have led to a lower median margin dose BED [50]. Unreported prognostic factors (number of lost to follow-up, systemic treatment and dosimetric parameters) could also explain this result. ”
- Table 1 concerning the risk of bias is very subjective. At least, authors must clarify the code used for the classification of the risk of bias.
Indeed, we tried to minimize the subjective character by assessing the risk of bias individually by following the ROBINS I tool and then by averaging the classification of each author.
- As a general comment, the tables are sometimes very difficult to read and interpret compared with the main text of the manuscript. For example, the authors talked about 279 patients selected in nine studies l.177 with the histologies. However, when we look at the table, the information concerning biopsies is given for five biopsies. Please clarify.
In the table, it is the data concerning the biopsies of the brain metastases in case of recurrence whereas in the text it is the histological type of the primary tumor. In order to make this difference more explicit we have changed the title of the relevant column in the table.
« Metastases biopsy avalaible »
Round 2
Reviewer 3 Report
I would like to thank the authors for having taken in account the suggestion made and clarified some points in the manuscript.